# Application of Autonomous Transportation Systems: Detection of a Potential Sub-Leasing Type of Carsharing

**Lide Yang [1], Jiemin Xie [2,*], Tuo Sun [1], Junxian Wu [3], Jinquan Hou [4] and Shuangjian Yang [5]**

1    Key Laboratory of Road and Traffic Engineering of the Ministry of Education, Tongji University, Shanghai 200092, China; yangld@tongji.edu.cn (L.Y.); suntuo@tongji.edu.cn (T.S.)
2    School of Intelligent Systems Engineering, Sun Yat-sen University, Guangzhou 510275, China
3    Anting Shanghai International Automobile City, Shanghai 215332, China; wujunxian@siac-sh.com
4    Gansu New Lugang Technology Co., Ltd., Gansu Highway Traffic Construction Group Co., Ltd., Lanzhou 730050, China; houjinquan7194@dingtalk.com
5    China Electric Vehicle Association, Shanghai 200051, China; yang.shuangjian@znwltech.com
*    Correspondence: xiejm28@mail.sysu.edu.cn

**Abstract:** Carsharing is regarded as an efficient way to reduce parking difficulty and road congestion, and the sub-leasing type of round-trip carsharing is an innovative way to enhance the efficiency of carsharing. Meanwhile, with the help of sub-leasing, the turnover rate of a parking lot increases, and the parking operator can earn a certain profit when the parking lot acts as a sub-leasing station. This study aims at extending the application of sub-leasing carsharing by suggesting an integrated framework of parking and sub-leasing based on the concept of autonomous transportation systems (ATSs), which are believed to be able to correctly sense demand, give proper reactions, and reduce operator workloads. A detection approach based on this ATS-based integrated framework is proposed to identify potential sub-leasing customers from the parking users, which is the main contribution of this study. Furthermore, this detection approach was tested using Shenyang parking data, and the parking lots with the greatest potential were analyzed in detail. The results show that 48.4% of parking activities had the potential to be transformed into sub-leasing activities. Of these potential activities, 52.7% were made by people who used a monthly payment scheme, and the promotion of sub-leasing should focus on these people.

**Keywords:** carsharing; sub-leasing; autonomous transportation systems; data-driven

## 1. Introduction

By the end of March 2022, China's car ownership reached 0.4 billion. Based on the general requirement, the number of parking spaces should be 1.3 times the number of cars. However, because of the limitation of land use, this requirement is barely met, especially in major cities in China. For example, according to the navigation data of Amap, when Shenzhen drivers arrive at their destinations, 11.6% of them fail to park at their destination and need to depart to search for an available parking space. With the increase in car ownership, parking has become one of the major transportation problems in downtown areas. Many strategies have been suggested to solve this problem, e.g., providing high-quality public transport support [1], pricing parking [2,3], reserving parking places [4], travel behavior educational programs [5], shared parking spaces [6], and parking path optimization [7]. One innovative solution is carsharing [8].

The majority of carsharing users are young people who are familiar with mobile apps [9], and carsharing has been growing fast and becoming more popular in recent years thanks to its combined characteristics from private and public transport [10]. Cars are owned by a carsharing company, and therefore people do not take ownership responsibilities but pay a small amount of money (compared to the expense of buying a car) for the use of a car, i.e., a large number of users share a small number of vehicles,

which is regarded as a sustainable solution to increase the usage of vehicles and therefore reduce car ownership, carbon emissions, and the need for land for parking [11–15]. In particular, electric carsharing can further reduce carbon emissions [16,17]. The details of recent research developments in carsharing can be found in the review of Nansubuga and Kowalkowski [18].

Carsharing can include more than one trip. For example, as shown in Figure 1, User X is a commuter who drives from home (Place A) to their workplace (Place B), parks the car somewhere close to the workplace, and later drives the car back home. This is round-trip carsharing. If carsharing only involves the trip to Place B or the trip to Place A, it is one-way carsharing. Feng et al. [19] compared these two types of carsharing and found that these two types of carsharing had different user patterns, while Le Vine et al. [20] found that compared to one-way carsharing, round-trip carsharing could more effectively reduce the total vehicle-miles of private cars and shared cars. Hence, these two types of carsharing should be studied separately. One-way carsharing is well-established and has been studied under different scenarios (i.e., carsharing with one operator or multiple operators) [21–23], whereas round-trip carsharing is more complicated and its potential is awaiting more exploration [24,25]. This study focuses on round-trip carsharing.

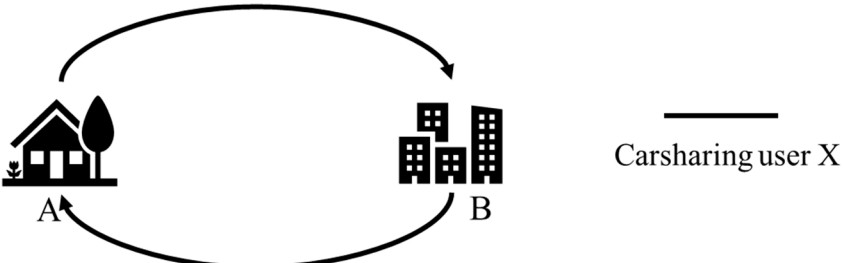

**Figure 1.** Trajectory of a sharing vehicle.

As previous studies [26] show, some factors have impacts on the further development of round-trip carsharing, including people's adoption, vehicle relocation efficiency, fleet size, pricing scheme, and parking. Parking is one of the major constraints, as it has been found that more than 90% of carsharing activities include a long stop [27]. For example, a user drives a car for a trip chain and generally prefers on-street parking because of convenience [28]. If only off-street parking is allowed, the user wants to park the car within an acceptable walking distance [29,30]. The tolerable walking distance is suggested to be 1.1 km [31]. Hence, a carsharing user would still face the problem of finding a nearby parking place. In addition, if the car is parked in a paid parking space, an expensive parking fee may be incurred. Some effective strategies have been suggested, like integrating parking-sharing into carsharing services [23] or introducing a sub-leasing type of round-trip carsharing [25].

As the work of Ziyadidegan et al. [25] introduces, the core idea of a sub-leasing car-sharing system is to reuse the parking time at intermediate stops so carsharing availability and accessibility increase and the fleet size of carsharing vehicles can be reduced. If User X shown in Figure 1 agrees to participate in the sub-leasing scheme, the preferred time frame for sub-leasing is given by User X in advance. For example, User X parks the car at Place B at 8 a.m. and leaves Place B at 6 p.m. Then, Place B becomes a temporary station, and carsharing User Y who travels between Place B and Place C (see Figure 2) can obtain the car after 8 a.m. and must return it before 6 p.m. If the parking time of User Y at Place C is sufficiently long, e.g., from 10 a.m. to 4 p.m., and User Y agrees to a further sub-leasing, User Z can take a round trip between Place C and Place D during the period from 10 a.m. to 4 p.m. Sub-leasing customers are required to follow certain rules, including being punctual, keeping the vehicle clean, and refilling gas when needed. Please note that vehicles could be privately owned by the "main" users, i.e., User X in Figure 2, or they could be part of a

fleet of a carsharing company and would be used solely by User X without the sub-leasing scheme considered in this study.

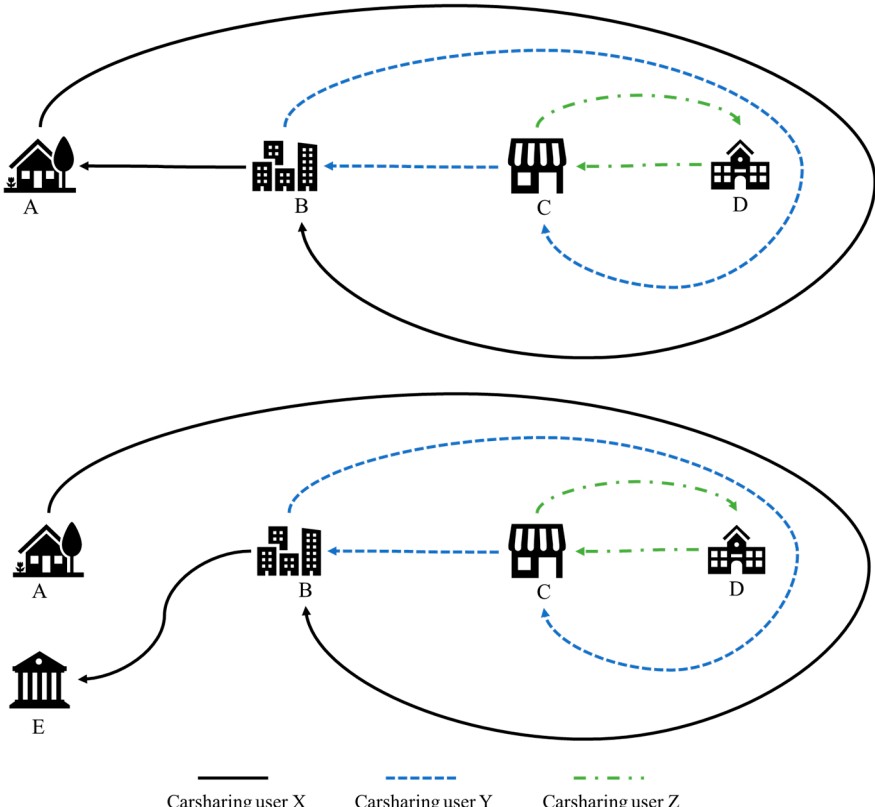

**Figure 2.** Trajectory of a sharing vehicle under the sub-leasing carsharing system.

This innovative approach can be further developed. First, the first user of the sharing vehicle cannot be a round-trip customer, but the followers (i.e., User Y and User Z) should be round-trip customers. For example, User X in Figure 2 can drive to Place E to return the car. Therefore, the number of targeted sub-leasing customers can be larger. Giving more flexibility would contribute to the improvement of carsharing services [32].

Second, parking operators and sub-leasing system operators can cooperate to reach a win–win situation. On the one hand, many cities in China are facing the problem of limited parking space, and the on-road search for a parking space is believed to be responsible for 30% of traffic congestion [6]. The sub-leasing system increases the availability and accessibility of parking spaces, i.e., parking spaces can serve more customers, which can contribute to the increase in parking profit. On the other hand, parking operators can analyze parking data to identify potential sub-leasing customers and help the sub-leasing system extend the market. For example, parking operators can identify which parking lots have private vehicles that can be replaced by sub-leasing carsharing vehicles, and then they can advertise in these parking lots to promote the concept of sub-leasing. Moreover, this can be beneficial to society, because people can enjoy better accessibility to carsharing vehicles and parking spaces, while sub-leasing customers can save parking fees because of a shorter parking time. Also, the reduction in vehicles leads to a reduction in carbon emissions.

However, few existing studies have considered the sub-leasing system [25]. Moreover, to the best of our knowledge, previous studies have not considered the cooperation of parking and sub-leasing, although the cooperation has the potential to improve carsharing and parking services. Our study exploits this gap by proposing a framework for the cooperation of parking and sub-leasing and then designing a detection approach to identify

potential parking lots with sufficient possible sub-leasing carsharing users for the promotion of sub-leasing.

Autonomous transportation systems (ATSs) will be introduced to ensure the efficiency of cooperation. Developing intelligent transportation systems (ITSs), ATSs automatically sense and manage diversified mobility demands with the help of new emerging technologies [33]. It is believed that with ATS, transport system operators and managers can be free from heavy daily workloads. For example, the prediction of the short-term future demand for carsharing is given by the Long Short-Term Memory structure rather than based on the operator experience [34]. In addition, modern information technology has improved carsharing schemes and made them more user-friendly [35]. Therefore, the cooperation of parking and sub-leasing should adopt ATS. To achieve the ATS-based framework, different functions, models, and methods should be properly designed. This study designs a detection method to help parking operators identify potential sub-leasing customers, while the other functions, models, and methods will be studied in the future.

This paper is organized as follows. The integrated framework of parking and sub-leasing and the sub-leasing customer-detection approach is presented in Section 2. Section 3 shows an empirical case of the detection approach, and the discussion is presented in Section 4. The conclusion is given in Section 5.

## 2. Methodology

This section presents the integrated framework of parking and sub-leasing in Section 2.1 and gives the details of the sub-leasing customer-detection approach in Section 2.2.

### 2.1. Integrated Framework of Parking and Sub-Leasing

With rapid technological development, the evolution of the transportation system continues, and ATS aims to describe this evolution. To face this new developing trend, the cooperation of parking and sub-leasing should be based on the concepts of ATS. Therefore, the integrated framework of parking and sub-leasing is built according to the service blueprint of ATS, which has been proposed in the work of You et al. [33], as shown in Figure 3. The integrated framework consists of four collaborative spaces, i.e., physical space, logical space, functional space, and technological space.

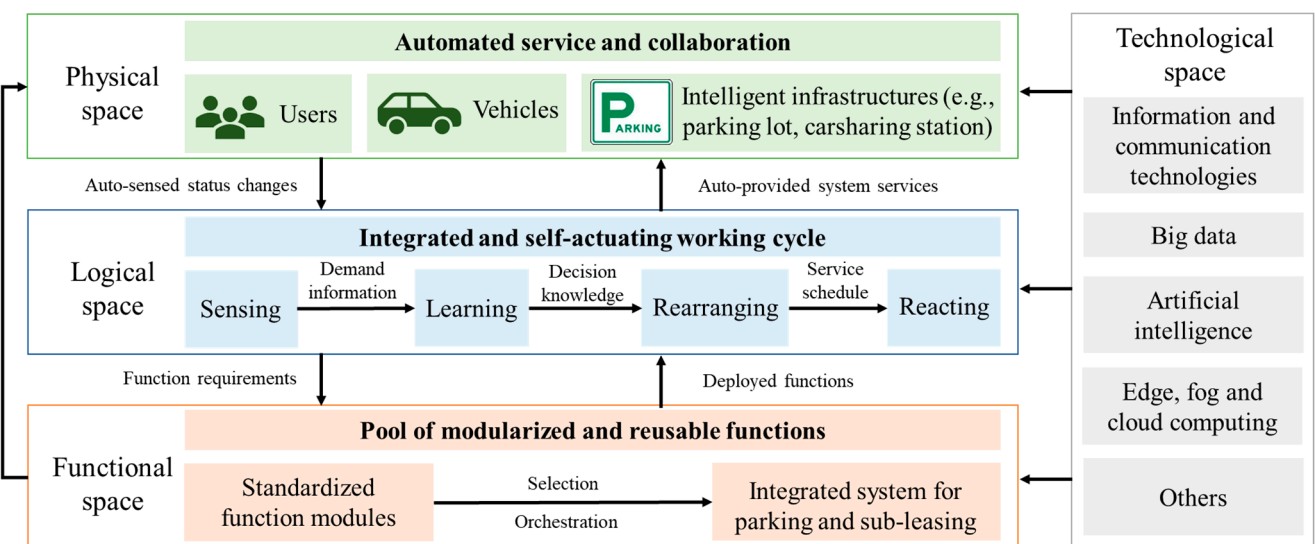

**Figure 3.** Integrated framework of parking and sub-leasing.

The physical space includes three types of participants that ATS handles in the physical world:

- Users who consume parking or sub-leasing carsharing services;

- Vehicles that are private or owned by carsharing companies; and
- Intelligent infrastructures that host and interact with users and vehicles, like a parking lot or a carsharing station where carsharing vehicles can park.

The logical space presents the internal working logic to show how ATS manages demand and supply. The logic is a self-actuating cycle with the following four steps:

- A sensing step, in which the changes in the physical work can be auto-sensed to identify demand information related to parking users and sub-leasing users;
- A learning step, in which demand information is processed to describe the present marker of sub-leasing, detect potential sub-leasing users, predict future status, and form decision knowledge;
- A rearranging step, in which the decision knowledge is internalized to design service schemes; and
- A reacting step, based on the generated schemes, in which parking and sub-leasing users are served, while potential sub-leasing users are encouraged to participate in sub-leasing carsharing.

In the functional space, the standardized function modules are built based on the functional requirements of the logical space, and then the selected function modules collaborate to form an integrated system of parking and sub-leasing for daily operation.

The technological space contains emerging technologies to support ATS-based parking and sub-leasing services. For example, big-data technologies help identify potential sub-leasing users.

This study mainly focuses on the logical space, which is introduced in detail in Section 2.2. An example is given in Section 3 to show how the suggested process works, while the core methods or technology involved in other spaces will be explored in the future.

### 2.2. Sub-Leasing Customer-Detection Approach

This section introduces the design of a potential sub-leasing customer-detection approach, with consideration given to cooperation with parking operators. The core idea of the detection approach follows the self-actuating logic cycle introduced in the ATS-based integrated framework of parking and sub-leasing. Moreover, the detection should be processed based on the updated status, so a dynamic periodic approach is proposed, as shown in Figure 4.

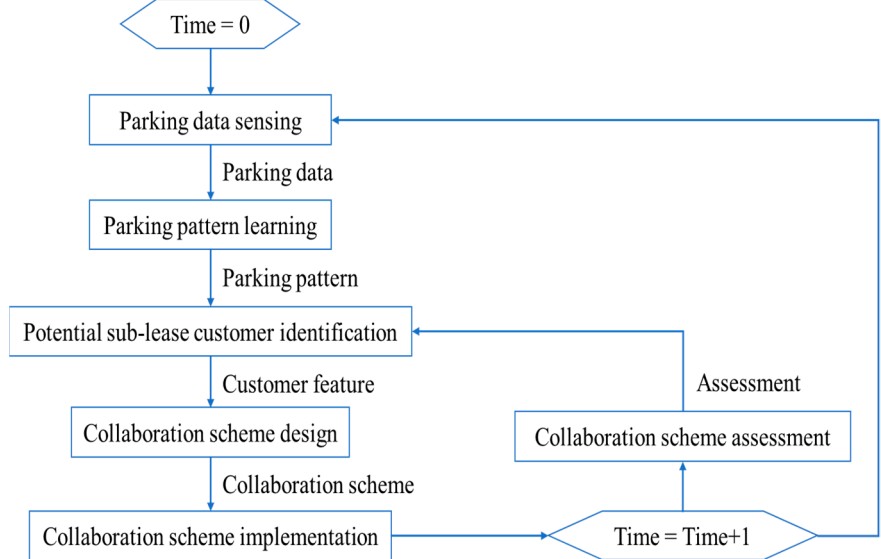

**Figure 4.** Sub-leasing customer-detection approach.

The length of a period is set based on the practical situation and, initially, the approach sets the time to be 0. The parking status is auto-sensed via the monitoring devices and the payment system, and then the related parking data are the input for parking pattern learning. The parking pattern is used to identify potential sub-leasing customers. The identification should be multiple layers, namely:

- Which parking lots are more likely to contain potential sub-leasing customers;
- When it is more likely that potential sub-leasing customers enter or leave their parking lots; and
- How potential sub-leasing customers pay for their parking fees.

This identification information can be used to design promotional strategies. The cost of advertising is highly variable. It is often based on the number of people who are expected to be reached, and tends to be higher at larger or busier parking lots. Moreover, the operators or investors generally prefer to maximize the benefits of sub-leasing promotion. Therefore, the main focus of the advertisement will be the parking lots that are more likely to serve potential sub-leasing customers. Moreover, if potential sub-leasing customers tend to use WeChat Pay [36] rather than Alipay [37], more advertisement investment should be given to WeChat Pay.

Then, the features of potential sub-leasing customers should be summarized because they are the foundation of the collaboration scheme design. The collaboration scheme design should consider:

- How to set parking fees and sub-leasing charges to attract people to participate in sub-leasing carsharing while balancing the profits of parking and sub-leasing carsharing companies; and
- How to effectively promote sub-leasing carsharing.

When the next period comes, not only will the updated parking status be checked, but the effect of the implemented collaboration scheme will also be assessed using the sensed demand and supply information. The assessment results and the updated parking pattern are combined to enhance the identification of potential sub-leasing customers and then improve the collaboration scheme.

This approach can adopt emerging techniques to achieve a high-quality solution. For example, in the collaboration scheme design, possible actions for collaboration are collected from journal papers and patents with the help of knowledge graph techniques in advance, and a pool of such actions is formed. Also, intelligent agents are built based on customer features and are used to simulate possible effects of the designed collaboration scheme before implementation.

## 3. Case Study

Because sub-leasing carsharing is a new concept, the practical application is in its infancy. Therefore, this study can only perform the first three steps of the sub-leasing customer-detection approach shown in Figure 4 (parking data sensing, parking pattern learning, and potential sub-leasing customer identification). Therefore, this study shows how to use the auto-collected parking data to identify potential sub-leasing customers in this section and suggests some possible actions for the fourth step (collaboration scheme design) in Section 4.

This section shows a case study in Shenyang, China. Shenyang [38] is the capital city of Liaoning Province, located in the northern part of China, with a total population of around 9 million people and a gross product of more than CNY 700 billion in 2021. In Shenyang, the number of registered vehicles was about 2.51 million at the end of 2021, and on average, there were 38.5 private vehicles per 100 families at the end of 2020.

Section 3.1 shows the general information on parking data, and Section 3.2 presents the detection of potential sub-leasing customers.

*3.1. Parking Data*

3.1.1. Data Collection Periods

The parking records were collected by the electric payment systems of 137 parking lots in Shenyang for two 7-day periods:

- Period 1: from 1 January 2021 to 7 January 2021; and
- Period 2: from 1 March 2021 to 7 March 2021.

The electric payment systems were designed by Neusoft, a Chinese IT service company, and installed at the entrance and exit of a parking lot. The electric payment systems generate parking records when a vehicle enters or leaves the parking lot, and then upload these parking records to a computer server. After privacy information was removed, the parking records were given to this study for further analysis.

All studied parking lots except for six offered information about the number of parking spaces. As shown in Figure 5, 57.3% of the studied parking lots had fewer than 100 parking spaces. The largest parking lot had 4117 parking spaces, and the smallest one had only 12 parking spaces, i.e., the dataset included different sizes of parking lots for analysis.

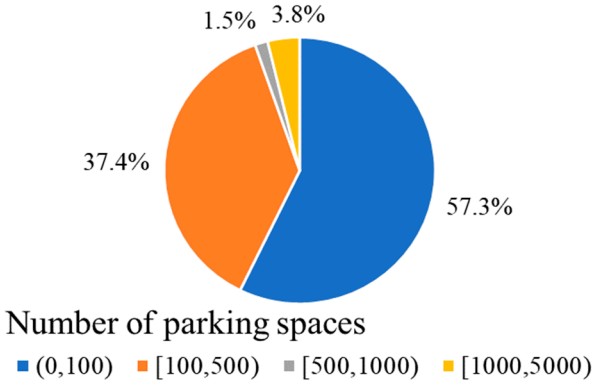

**Figure 5.** Distribution of the size of a parking lot.

3.1.2. Data Structure

A parking record of a vehicle includes the following information.

- Car ID: The China vehicle ID generally shows a Chinese character of the abbreviation of the province in which the vehicle is registered, a Latin character to represent the city in which the vehicle is registered, and five numbers, as shown in Figure 6. The first Chinese character refers to Liaoning Province, and "A" refers to Shenyang. To protect the privacy, only the Chinese character and the Latin character of the obtained car ID were given to us for analysis.

**辽A 00000**

**Figure 6.** Example of a vehicle ID (The first Chinese character refers to the province in which the vehicle is registered, and, in this figure, the first Chinese character refers to Liaoning Province).

- Way of payment: Because the parking records were collected by the electric payment systems, there was no record in which the parking fee was paid by cash. Drivers can pay a parking fee via WeChat Pay, Alipay, and electronic toll collection (ETC). When people use WeChat Pay and Alipay, they need to use the respective mobile app to scan the QR code to pay. However, if the ETC is used, an ETC card is installed in an automated radio transponder device in a vehicle, and a roadside toll-reader device detects the entrance and the exit of the vehicle via the connection with the automated radio transponder device, and then the user is charged. Moreover, if a driver adopts

the ETC, the driver can pay a fee after each parking session, or the driver can be a VIP customer who uses a monthly payment scheme. Therefore, there are four types of parking customers, namely WeChat Pay, Alipay, General ETC (GETC), and VIP.

- Enter time: The enter time of a vehicle is the time when the vehicle enters the parking lot, including information about the year, month, day, hour, minute, and second.
- Leave time: The leave time of a vehicle is the time when the vehicle leaves the parking lot, including information about year, month, day, hour, minute, and second.
- ID of the parking lot: To protect privacy, the locations and names of the studied parking lots were not given. However, the parking operator provided pseudo codes for these parking lots, so parking records of different parking lots can be distinguished.

### 3.1.3. Data Cleaning

In total, 467,266 parking records were obtained. Due to machine errors, some records were unreliable. Therefore, data cleaning was needed. A pre-process was established using a simple principle: if the parking duration was negative or less than 5 min, this parking record was regarded to be incorrect and then deleted. The parking duration is calculated as follows.

$$\text{Parking duration} = \text{Leave time} - \text{Enter time.} \tag{1}$$

After data cleaning, the number of parking records fell to 416,673, i.e., 89.2% of parking records were logically reliable in terms of parking duration and kept for further analysis. This process might cause some potential bias because some records were deleted. However, in this case, more than 90% of parking records were kept and the main focus of further analysis was parking with a sufficient long parking duration for sub-leasing carsharing, so potential biases would have few impacts on the analysis.

### 3.1.4. Basic Characteristics of Parking Customers

In this dataset, 2.5% of parking vehicles came from different provinces of China. More than 25% of parking vehicles had a Jilin license plate, 12.4% of parking vehicles had a Heilongjiang license plate, and 11.0% of parking vehicles had a Beijing license plate because Beijing, Jilin, and Heilongjiang are close to Liaoning Province. Except for the customers with a Guizhou license plate, the customers with a non-Liaoning license plate were more likely to use WeChat Pay. For example, 34.3% of parking vehicles with a Jilin license plate used WeChat Pay, whereas 24.0% of parking vehicles with a Liaoning license plate used WeChat Pay.

A total of 97.5% of data records were related to parking vehicles whose license plates belong to Liaoning Province. A total of 93.1% of parking vehicles with a Liaoning license plate had a license plate from Shenyang. For other parking vehicles with a Liaoning license plate, 31.7% of them had Liaoyang license plates, and 15.1% of them had Huludao license plates, as shown in Figure 7.

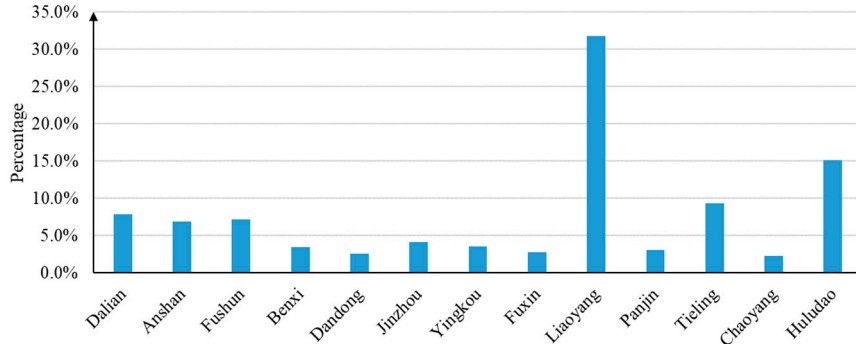

**Figure 7.** Distribution of parking vehicles with a Liaoning license plate.

Therefore, the data mainly reflects the parking behaviors of customers with a Shenyang license plate who tend to be Shenyang residents or who live in Shenyang for a relatively long period of the year. This probably explains why 73.5% of data records were made by VIP customers (Figure 8). If drivers did not adopt the monthly payment via ETC, they were less likely to use ETC to pay parking fees. Moreover, 24.2% of data records were made by WeChat Pay customers, whereas only 0.3% of data records were made by Alipay customers, i.e., WeChat Pay seemed to have a larger market share than Alipay in Shenyang.

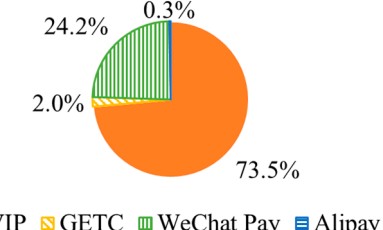

**Figure 8.** Distribution of parking records in terms of payment way.

Considering January has a festival (i.e., New Year Festival), data were separated by month. For January, the entering and leaving patterns are shown in Figures 9 and 10.

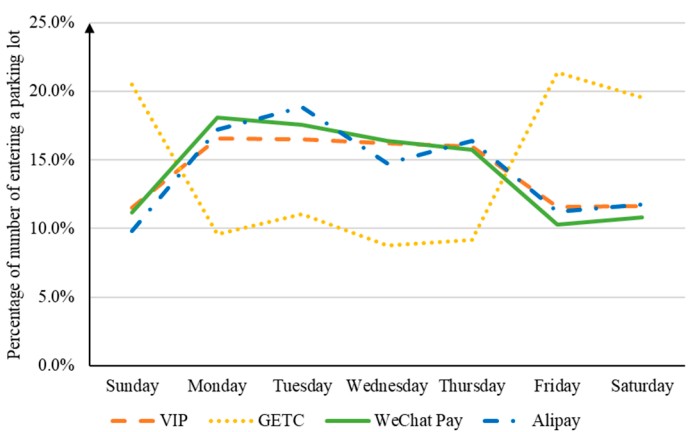

**Figure 9.** Distribution of parking vehicles in terms of entering day in January 2021.

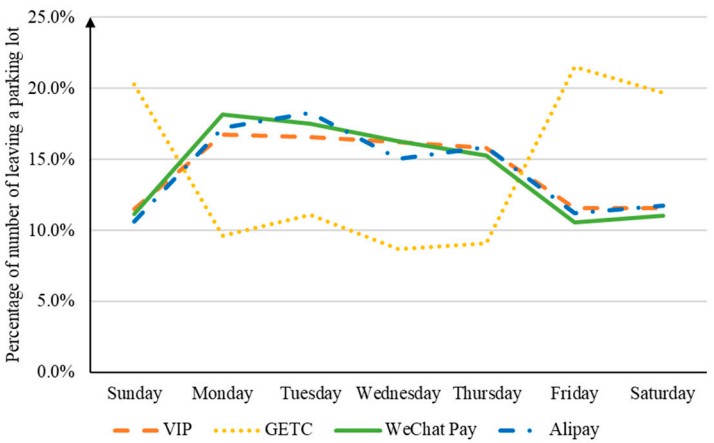

**Figure 10.** Distribution of parking vehicles in terms of leaving day in January 2021.

VIP customers shared a similar pattern of entering a parking lot with customers who paid parking fees by WeChat Pay and Alipay. The pattern showed that similar entering

rates appeared from Monday to Thursday, and a fall occurred on Friday. GETC customers had a different pattern and a peak on Friday, Saturday, and Sunday. 1 January 2021 was a Friday and the holiday for the New Year was three days long. People who installed an ETC system on their vehicles but did not adopt the monthly payment seemed not to be regular drivers. However, they seemed to increase driving trips during the New Year holiday, whereas other customers reacted differently and reduced car trips. Without the holiday effect, GETC customers had similar patterns to other customers, as shown in Figures 11 and 12, which present the entering and leaving patterns in March.

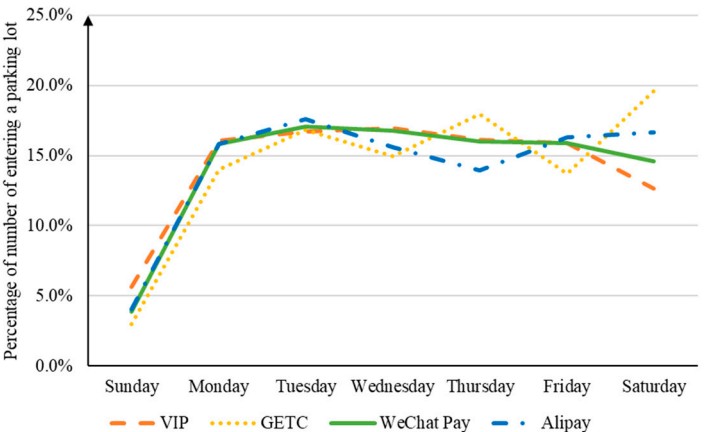

**Figure 11.** Distribution of parking vehicles in terms of entering day in March 2021.

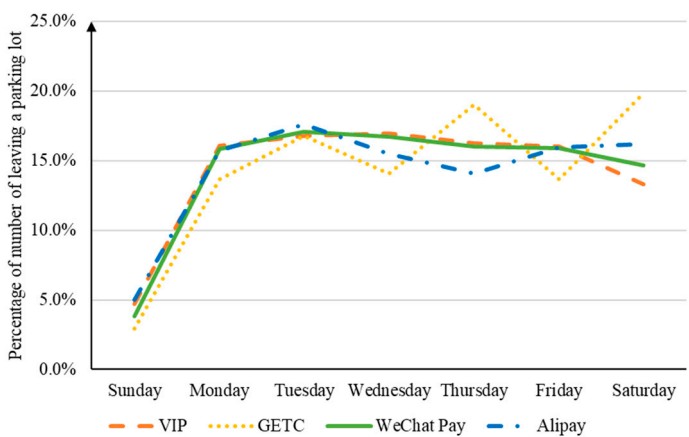

**Figure 12.** Distribution of parking vehicles in terms of leaving day in March 2021.

In all, most entering and leaving activities happened during the weekdays, indicating that commuter trips could be the major contributors to parking. If it is a normal weekend, parking is active on Saturday but inactive on Sunday. People may go out for fun with family and friends on Saturday, such as going shopping or visiting a tourist interest. However, people would like to take a rest at home on Sunday because the next day is a working day.

A total of 95.1% of parking activities were completed within one day (Figure 13), whereas other parking activities had a long parking duration which should be considered transformable into carsharing activities to reduce the waste of vehicles and parking spaces. Furthermore, one-day parking activities also had the potential to be transformed. As shown in Figure 14, 61.1% of parking activities had a relatively long parking duration, i.e., more than 1 h. According to the previous studies [25], the critical period, i.e., the minimum time interval, to guarantee the success of the sub-leasing carsharing system, is 1 h, i.e., if a vehicle parks for a period that is longer than the critical period ($T^*$), the vehicle can be

leased to another driver. This study also adopts this setting to identify potential sub-leasing activities. The next section discusses this in more detail.

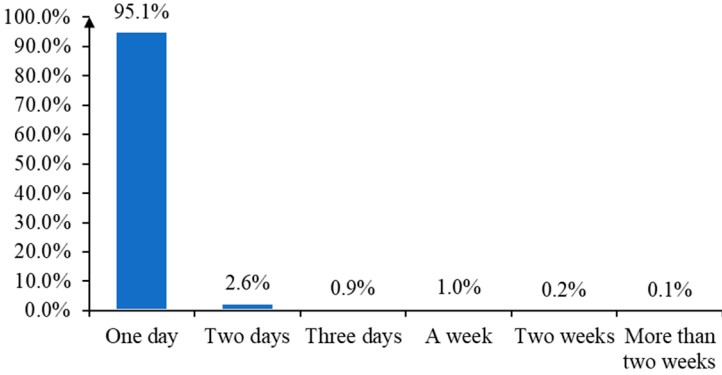

**Figure 13.** Distribution of parking vehicles in terms of parking duration.

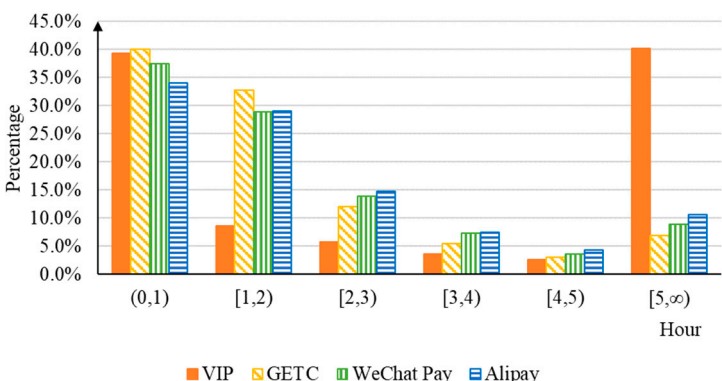

**Figure 14.** Distribution of parking vehicles in terms of parking duration with consideration of different payment methods.

In conclusion, considering the distributions in Figures 8 and 14, VIP customers and WeChat Pay customers were the majority and had a relatively long parking duration; therefore, they should be the main targets of sub-leasing promotion. Furthermore, a relatively stable parking pattern of VIP customers and WeChat Pay customers during the weekdays is shown in Figures 11 and 12, indicating that the supply level of potential sub-leasing cars is relatively similar during the weekday period.

### 3.2. Detection of Potential Sub-Leasing Customers

#### 3.2.1. Detection Process

Personal privacy has become a major concern in society, so directly sending an advertisement may cause people to become averse to the advertised product. Therefore, this study considers advertising in some specific parking lots that have sufficient potential sub-leasing customers and the possibility of becoming a sub-leasing station.

Whether a parking lot has sufficient potential sub-leasing customers can be checked by whether it has sufficient potential parking activities whose parking duration is not less than $T^*$. If the average number of potential parking activities in a day in a parking lot is not less than $P^{\text{Time}}$, this parking lot can be considered to have sufficient potential parking activity.

As for the possibility of it becoming a sub-leasing station, a parking lot should be popular. A parking lot may have many vehicles with very long parking periods, but may be located far away from the city center, which few people visit. In this case, no one will go there to pick up a sub-leasing vehicle. This study uses the number of leavings in a day to measure the popularity of a parking lot. If the average number of leavings in a day in

a parking lot is not less than $P^{\text{Leave}}$, this parking lot can be considered to have sufficient popularity.

Overall, a parking lot without sufficient potential parking activity and popularity should not be selected as a target for sub-leasing carsharing promotion. If selected, the investment is likely wasted. Therefore, with consideration given to the data in this example, a detection process has been designed as shown in Table 1, and coded in C#. The detection process can be run by a computer automatically once the sensed data are ready.

**Table 1.** Detection process.

| Detection Process |
|---|
| Input: Sensed data |
| Output: A list of parking lots with their characteristics and the relative business strategy implications |
| Step 1 : Input the sensed data. |
| Step 2 : Process data cleaning. |
| Step 3 : Divide parking records into different sets according to parking lots. |
| Step 4 : Select the parking lots that have potential parking activities according to parking duration and form the first set of parking lots. |
| Step 5 : In the first set, select the parking lots whose average number of potential parking activities in a day has a non-decreasing trend and form the sec ond set of parking lots. |
| Step 6 : In the sec ond set, select the parking lots whose average number of leavings in a day has a non-decreasing trend and form the third set of parking lots. |
| Step 7 : Calculate the descending orders of parking lots in terms of the average number of potential parking activities in a day in January $(O_1^P)$, the average number of potential parking activities in a day in March $(O_3^P)$, the average number of leavings in a day in January $(O_1^L)$, and the average number of leavings in a day in March $(O_3^L)$, i.e., if orders $O_1^P$, $O_3^P$, $O_1^L$, and $O_3^L$ of a parking lot are smaller, the parking lot has more potential parking activities. |
| Step 8 : Calculate the weighted sum $(S)$ as follows : |
| $$S = \beta_1^P O_1^P + \beta_3^P O_3^P + \beta_1^L O_1^L + \beta_3^L O_3^L \qquad (2)$$ |
| Step 9 : For a parking lot, if $S$ is the top $P^{\text{Order}}\%$ among the parking lots in the third set, it is selected and put into the fourth set. |
| Step 10: Analyze the characteristics of the parking lots in the fourth set. |
| Step 11: Design business strategy implications based on the analysis. |
| Step 12: Output the fourth set and business strategy implications. |

In this study, Step 2 deletes the records with a logical problem, as mentioned in Section 3.1.3.

For Step 4, this study checked both months, January and March. $N_1^P$ and $N_3^P$ refer to the average potential parking activities per day in January and March, respectively, while $N_1^L$ and $N_3^L$ refer to the average leavings in a day in January and March, respectively. If there were potential parking activities in only one month, the supply of potential parking activities might not be stable. Therefore, the process selected the parking lots whose average potential parking activities per day in both months were not less than 1.

For Steps 5 and 6, this study can only analyze the trend between January 2021 and March 2021. More data in different months can show a better developing trend. A non-decreasing trend in the average number of potential parking activities in a day (i.e., $N_3^P - N_1^P \geq 0$) may indicate a non-decreasing number of potential sub-leasing users, whereas a non-decreasing average number of leavings in a day (i.e., $N_3^L - N_1^L \geq 0$) may indicate a non-decreasing popularity of parking. Therefore, parking lots with these tendencies should be selected.

Steps 7–9 have two advantages. First, if the operators can set values for $P^{\text{Time}}$ and $P^{\text{Leave}}$, it would be easier to determine the targeted parking lots. However, it is difficult to set the values of $P^{\text{Time}}$ and $P^{\text{Leave}}$ properly at the initial stage. Selecting the parking lots that rank in the first level would be a reasonable alternative. Second, the investment in marketing is always limited, and therefore the initial promotion may be set in a limited number of parking lots. This proposed process can handle this and select the initial targets according to limitations. However, due to the limited data, only two months of data were

used, which may not perfectly represent the characteristics of parking lots. If possible, the operators should try to include as much data as possible. Furthermore, regarding the weighting method used in Step 8, improper weights could result in biased detection results, so weights should be set with careful consideration of local features.

In Step 10, the analysis can include entering times, leaving times, and payment ways, as mentioned in Section 2.2.

In this analysis, $T^*$ was set to be 1 h based on the previous studies [25], while $P^{\text{Order}}$ was set to 15 because the top 15% is generally regarded to be sufficiently good. In addition, $\beta_1^P = \beta_3^P = \beta_1^L = \beta_3^L = 0.25$. The aforementioned parameters can be adjusted based on the assessment result after the implementation of the promotional scheme, so parking lots with a higher potential can be found.

### 3.2.2. Detection Results

Table 2 shows the number of parking lots in each set during the detection process. Finally, 14 parking lots were selected as the targeted ones with relatively more potential sub-leasing customers, and Tables 3 and 4 present the information on these parking lots. The last four columns of Table 3 show the normalization values of $N_1^P$, $N_3^P$, $N_1^L$, and $N_3^L$. The normalization value was calculated by dividing the initial value by the number of parking spaces. Generally, a small parking lot can only serve a limited number of vehicles and therefore it is more likely to suffer from parking difficulty. If sub-leasing carsharing is adopted in a small parking lot, the parking difficulty may be reduced. However, the promotion of sub-leasing carsharing in this parking lot may not cover many people, and it is not sufficiently profitable at the earlier stage of sub-leasing carsharing development. As Table 3 shows, the selected high-potential parking lots generally have more than 100 parking spaces. Therefore, the selection of parking lots at the earlier stage of sub-leasing carsharing development should consider larger parking lots.

**Table 2.** Number of parking lots in each set.

|  | The Number of Parking Lots |
| --- | --- |
| First set | 104 |
| Second set | 99 |
| Third set | 97 |
| Fourth set | 14 |

**Table 3.** Parking lots in the fourth set (1).

| Parking ID | Number of Parking Spaces | $N_1^P$ | $N_3^P$ | $N_1^L$ | $N_3^L$ | $\tilde{N}_1^P$ | $\tilde{N}_3^P$ | $\tilde{N}_1^L$ | $\tilde{N}_3^L$ |
| --- | --- | --- | --- | --- | --- | --- | --- | --- | --- |
| 1001 | 208 | 929.57 | 1549.00 | 2317.71 | 2803.71 | 4.47 | 7.45 | 11.14 | 13.48 |
| 5033 | 550 | 462.14 | 1188.71 | 1228.71 | 2181.86 | 0.84 | 2.16 | 2.23 | 3.97 |
| 4008 | 166 | 496.00 | 784.43 | 990.43 | 1318.71 | 2.99 | 4.73 | 5.97 | 7.94 |
| 5061 | 800 | 739.14 | 818.00 | 915.86 | 984.00 | 0.92 | 1.02 | 1.14 | 1.23 |
| 3006 | 120 | 497.00 | 574.43 | 735.57 | 803.57 | 4.14 | 4.79 | 6.13 | 6.70 |
| 4005 | 206 | 362.57 | 502.14 | 652.43 | 777.57 | 1.76 | 2.44 | 3.17 | 3.77 |
| 4003 | 347 | 239.43 | 321.14 | 758.71 | 1035.29 | 0.69 | 0.93 | 2.19 | 2.98 |
| 2004 | 93 | 352.00 | 533.57 | 595.71 | 732.86 | 3.78 | 5.74 | 6.41 | 7.88 |
| 3008 | 189 | 233.57 | 363.29 | 631.14 | 840.57 | 1.24 | 1.92 | 3.34 | 4.45 |
| 5045 | 1208 | 385.00 | 618.14 | 420.00 | 688.86 | 0.32 | 0.51 | 0.35 | 0.57 |
| 4004 | 1005 | 408.86 | 529.14 | 471.14 | 613.00 | 0.41 | 0.53 | 0.47 | 0.61 |
| 2007 | 121 | 190.29 | 973.00 | 250.00 | 1237.00 | 1.57 | 8.04 | 2.07 | 10.22 |
| 5007 | 110 | 286.00 | 428.86 | 483.00 | 722.43 | 2.60 | 3.90 | 4.39 | 6.57 |
| 6004 | 353 | 239.00 | 323.14 | 563.29 | 738.43 | 0.68 | 0.92 | 1.60 | 2.09 |

**Table 4.** Parking lots in the fourth set (2).

| Parking ID | $O_1^P$ | $O_3^P$ | $O_1^L$ | $O_3^L$ | $S$ |
|---|---|---|---|---|---|
| 1001 | 1 | 1 | 1 | 1 | 1.00 |
| 5033 | 5 | 2 | 2 | 2 | 2.75 |
| 4008 | 4 | 5 | 3 | 3 | 3.75 |
| 5061 | 2 | 4 | 4 | 7 | 4.25 |
| 3006 | 3 | 8 | 6 | 10 | 6.75 |
| 4005 | 8 | 11 | 7 | 11 | 9.25 |
| 4003 | 11 | 18 | 5 | 6 | 10.00 |
| 2004 | 9 | 9 | 9 | 13 | 10.00 |
| 3008 | 13 | 15 | 8 | 9 | 11.25 |
| 5045 | 7 | 7 | 16 | 16 | 11.50 |
| 4004 | 6 | 10 | 13 | 17 | 11.50 |
| 2007 | 17 | 3 | 24 | 4 | 12.00 |
| 5007 | 10 | 13 | 12 | 15 | 12.50 |
| 6004 | 12 | 17 | 10 | 12 | 12.75 |

## 4. Discussion

### 4.1. Analysis of Top 1 Parking Lot

Without loss of generality, the top parking lot with 35,850 records was selected to analyze its basic characteristics. In this parking lot, 97.8% of parking vehicles came from Liaoning Province, and 95.7% of parking vehicles with a Liaoning license plate had a license plate of Shenyang. A total of 97.5% of parking activities were completed within one day, as shown in Figure 15. It is consistent with the previous finding that less than 3% of parking activities were longer than one day [28]. As shown in Figure 16, 48.4% of parking activities had a relatively long parking duration, which was more than 1 h, and these activities had the potential to be sub-leasing activities based on the previous studies [25].

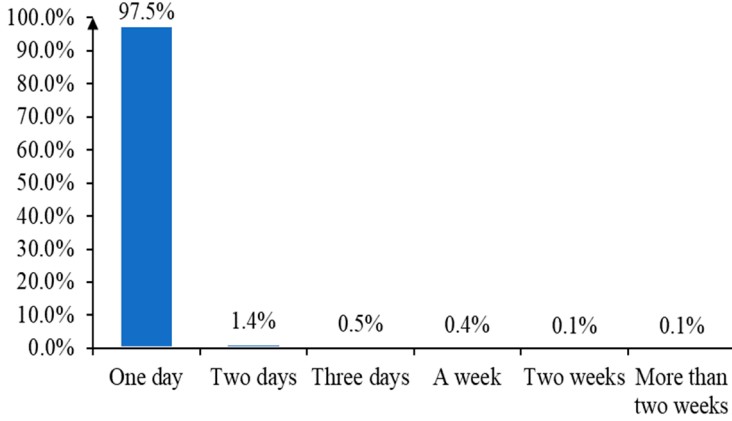

**Figure 15.** Distribution of parking vehicles in terms of parking duration in the top parking lot.

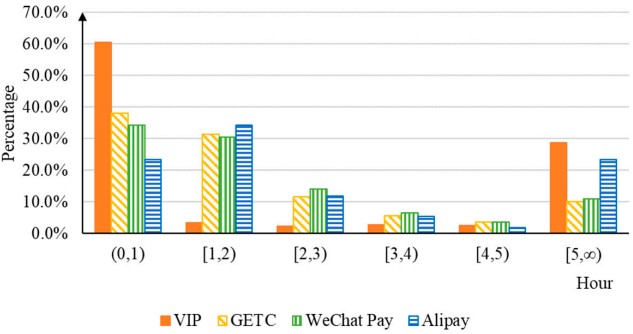

**Figure 16.** Distribution of parking vehicles in terms of parking duration in the top parking lot.

For potential parking activities, 52.7% of data records were made by VIP customers (Figure 17), and 35.8% of data records were made by WeChat Pay customers. Hence, VIP and WeChat Pay customers should be the main target of the sub-leasing carsharing promotional scheme. Moreover, as shown in Figure 18, WeChat Pay customers' parking durations tended to be shorter than those of VIP customers. Therefore, WeChat Pay customers would be the target of a simple sub-leasing activity, whereas VIP customers could be the target of a complex sub-leasing activity. In a simple sub-leasing activity, the vehicle is sub-leased to one user to have a round trip going to a relatively closed place. In addition, in a complex sub-leasing activity, the vehicle could be sub-leased to more than one user or a user to visit a place much further away.

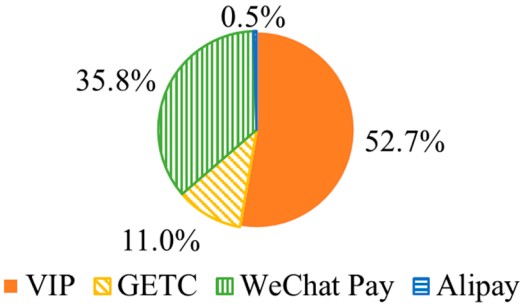

**Figure 17.** Distribution of potential parking activities in terms of payment way in the top parking lot.

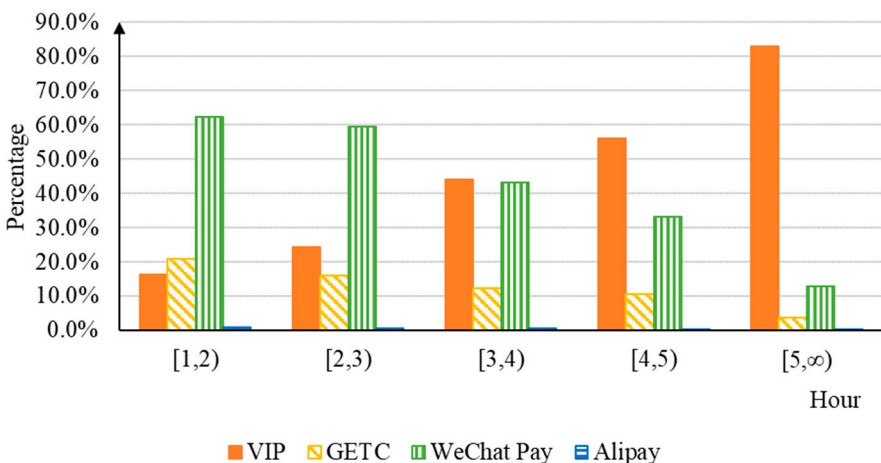

**Figure 18.** Distribution of potential parking activities in terms of payment way and parking duration in the top parking lot.

The distributions of entering and leaving times of potential parking activities of VIP and WeChat Pay customers are presented in Figures 19 and 20.

For VIP customers, an obvious peak was found when the entering time was within the period from 06:00 to 08:00, which was consistent with the previous finding in a southern city of China [6], i.e., the parking demand in the early morning was found to be highest, but the peak in Shenyang seemed to be earlier than that in a southern city of China because the working schedule in northern cities (e.g., Shenyang) is generally earlier than that in southern cities. Another obvious peak was found when the leaving time was within the period from 16:00 to 18:00. A total of 36.0% of VIP customers seemed to have a commuter pattern in which a vehicle enters the parking lot between 6:00 and 9:00 and leaves between 16:00 and 19:00. This pattern was similar to previous findings [3,15]. A commuter usually drives a car to the workplace and parks the car near the workplace in the morning. After work, the commuter leaves the parking lot in the evening. For such potential sub-leasing

customers, a complex sub-leasing activity can be considered because their parking duration is relatively long.

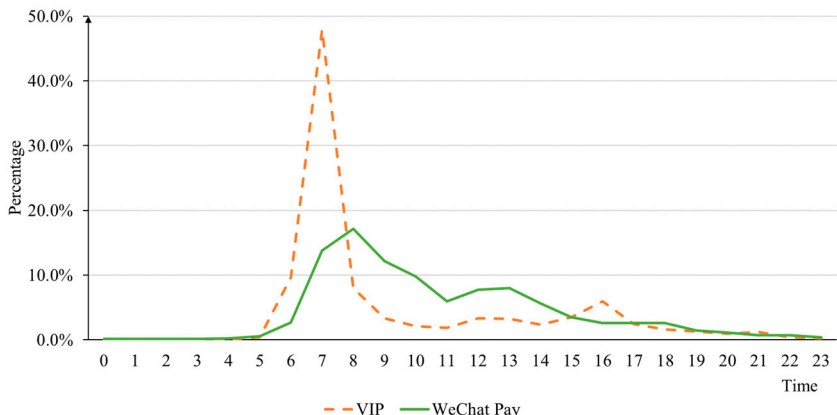

**Figure 19.** Distribution of potential parking activities in terms of entering time in the top parking lot.

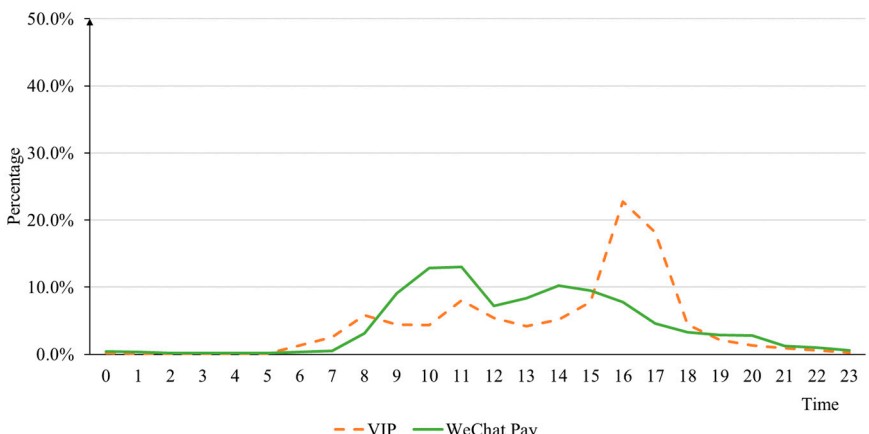

**Figure 20.** Distribution of potential parking activities in terms of leaving time in the top parking lot.

However, this pattern was not found among WeChat Pay customers. Although most entering activities of WeChat Pay customers happened in the morning, it was distributed more evenly, compared to those of VIP customers. Additionally, the leaving of WeChat Pay customers was more likely to happen between 09:00 and 15:00, earlier than that of VIP customers. With consideration given to these patterns, the advertisement can be shown during these periods to maximize the impact of advertisement investment.

### 4.2. Business Strategy Implications

Based on the analysis in the previous sections, the possible business strategy implications are summarized as follows.

- The differences between weekdays and weekends in terms of car parking indicate that potential demands for sub-leasing carsharing on weekdays could be different from those at weekends, and specific schemes can be designed. For Shenyang, compared with the weekend, weekdays had more parking activities, indicating that the parking need was higher, and the parking difficulty tended to be more serious. A cheaper charge scheme for sub-leasing carsharing might attract more customers and then reduce parking difficulty.
- Parking lots with relatively more potential parking activities and high popularity should be selected as the targeted parking lots.

- VIP and WeChat Pay customers should be the main target of the sub-leasing carsharing promotional scheme. The promotion can be sent via the platforms of ETC and WeChat Pay to customers who park their cars at the targeted parking lots. However, the methods for sending ads should be different because different payment methods are used for VIP and WeChat Pay customers. WeChat Pay customers need to use the WeChat app to scan a QR code to pay. However, for VIP customers, automated radio transponder devices installed in their vehicles communicate with roadside toll-reader devices to detect entrances and exits of vehicles, and then the ETC system manages the charges of VIP customers without manual operation. Therefore, for WeChat Pay customers, promotional ads should be sent after WeChat Pay customers scan QR codes, whereas, for VIP customers, the promotional ads should be sent with monthly bills.
- VIP customers should be encouraged to try a complex sub-leasing activity. In the advertisement for VIP customers, the monetary reward of sub-leasing activities should be the main focus. Because their parking duration is relatively longer and the parking fee is therefore relatively higher, punctuality might not be the main concern, but a saving in parking fee might awaken their interest.
- WeChat Pay customers should be encouraged to join a simple sub-leasing activity. In the advertisement for WeChat Pay customers (e.g., the payment page or interface showing the advertisement for sub-leasing), the reliability of sub-leasing activities should be emphasized. Because their parking duration is relatively shorter, they would like to know that their vehicle will return on time.
- During the period from 07:00 to 15:00, the chance of showing sub-leasing advertisements on the WeChat payment page should be increased, because the people using WeChat Pay in this period have a higher potential to participate in a sub-leasing activity.
- The promotional scheme can consider using coupons, discounts, and packages which can attract more carsharing activities [24].

## 5. Conclusions

There are two contributions to this study. First, this study suggests a new integrated framework for the collaboration of sub-leasing carsharing companies and parking companies, which is based on ATS with emerging technology. Based on the logical space of the integrated framework, a detection approach is proposed to identify potential sub-leasing customers from parking users. Second, to show the detection approach, a case study with Shenyang parking data has been performed to find the parking lots with relatively more potential sub-leasing customers. Some interesting findings related to sub-leasing carsharing have been found and the possible business strategy implications have been given.

There are three major findings.

First, the results showed that the parking patterns on weekdays and weekends were different, indicating that sub-leasing carsharing activities on weekdays and weekends would have different commercial patterns. The operators should adopt diverse operation approaches to meet customer needs on weekdays and weekends. Generally, travel patterns on weekdays and weekends vary no matter which cities are considered, so this implication can be applied to other cases.

Second, VIP and WeChat Pay customers in Shenyang were more likely to participate in sub-leasing carsharing, because their parking times were sufficiently long for at least one sub-leasing activity. In addition, due to the heterogeneity of the parking patterns, different business strategies should be designed to attract VIP and WeChat Pay customers, respectively. In other cities, WeChat Pay could be less popular than Alipay, so in this case WeChat Pay customers cannot be regarded as the major target of promotion. However, the payment methods of WeChat Pay and Alipay are similar, and therefore similar promotional schemes can be used to attract Alipay customers. In addition, VIP customers who generally have a high parking demand and then purchase a long-term parking scheme may share a

similar parking pattern, no matter which cities are considered, i.e., VIP customers generally have a high potential to be sub-leasing carsharing customers.

Third, some of the findings based on Shenyang parking data can be applied to other cities (e.g., the parking demand in the early morning was found to be highest), as the comparison with the previous literature shows. However, at this stage, there is no data collected in other cities to compare the similarities and differences among different cities. In addition, it is a possible analysis direction to collect data from other cities for further comparison.

Moreover, the demand side (i.e., customer demographics, behavior, preferences related to sub-leasing carsharing, and whether there are people who would use short-term sub-leasing of vehicles during the daytime) is not considered, which is a key limitation of this study. On the one hand, it is not easy to collect demand data. In addition, sub-leasing carsharing is in its infancy, so further data for the interaction between customers and operators is not available. On the other hand, a full evaluation of the daily imbalance in mobility (i.e., more vehicles are used during rush hour, and tidal travel flows exist) is a much more complex question and it is out of the scope of the current work. Therefore, this study mainly considers the supply side (i.e., the availability of vehicles), and discusses the promotion and advertisement of services from the supply side. In the future, the demand side should be included, and more relative issues should be considered, including customer demographics, behavior, preferences related to sub-leasing carsharing, and the impact factors on customer sub-leasing carsharing behavior (e.g., payment methods), via a series of stated-preference and revealed-preference surveys. Also, a further simulation game will be designed as a pilot test for sub-leasing carsharing.

**Author Contributions:** Conceptualization, L.Y. and J.X.; methodology, J.X.; validation, T.S.; data curation, L.Y., J.W. and J.H.; writing—original draft preparation, L.Y. and J.X.; writing—review and editing, S.Y.; funding acquisition, T.S. All authors have read and agreed to the published version of the manuscript.

**Funding:** This work was supported by the Shanghai Science and Technology Innovation Action Plan Project (projects no. 22YF1452700, 20DZ1202805, and 22002400100), Gansu Science and Technology Major Project (project no. 22ZD6GA010).

**Institutional Review Board Statement:** Not applicable.

**Informed Consent Statement:** Not applicable.

**Data Availability Statement:** The data that support the findings of this study were provided by Neusoft. Restrictions apply to the availability of these data, which were used under license for this study.

**Acknowledgments:** The authors express their sincere gratitude to Neusoft for the provision of the data used in this research.

**Conflicts of Interest:** The authors declare no conflict of interest.

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
