# Peer review of "Application of Autonomous Transportation Systems: Detection of a Potential Sub-Leasing Type of Carsharing"

_sustainability, doi:10.3390/su151914220_

Round 1

Reviewer 1 Report

This study proposes an integrated framework of parking and sub-leasing to extend the application of carsharing. The main concerns are listed as follows:

1. The contribution of the paper is not clarified in Abstract. Adding numerical results is useful to improve it.

2. The previous literatures are not sufficient. A more comprehensive literature review can clarify the contributions for the paper.

3. The core methods or technology involved in the proposed framework are not clear.

4. A clear description regarding all steps should be given before giving the description that this study can only perform the first three steps, e.g., in Section 3.

5. The contributions and novelty of the study are not clear in the conclusion. 

Author Response

We thank the reviewer for the constructive comments. We have revised our manuscript accordingly. For your convenience, we explain how we have addressed the comments in the attached response. The comments have been marked in italics and our reply is in regular font, while the revised tests in the manuscript are marked in blue.

Reviewer 2 Report

The study delves into sub-lease carsharing, acknowledging it's a relatively new concept, and explores the initial steps of identifying potential sub-lease customers. It employs a case study from Shenyang, China, to illustrate its points.

The manuscript is well-written and interesting to the journal's readers, so it could be publishable if the recommendations described below were followed.

General comments

Methodology

Parking Data

The paper does well in introducing the parking data used for analysis. However, some information on data sources and collection methods could enhance transparency and credibility. Furthermore, while discussing parking lot sizes, it would be beneficial to understand how these sizes impact sub-lease carsharing feasibility.

Data Structure

The explanation of the data structure is clear, but more details about the payment methods and their implications for sub-lease promotions would be valuable. For example, how do WeChat Pay, Alipay, ETC affect customer behaviors regarding sub-lease carsharing?

Data Cleaning

The data-cleaning process is appropriately described, but the paper lacks insight into potential biases introduced during this phase and how they might affect the results.

Basic Characteristics of Parking Customers

This section provides essential insights into parking customers, but it could benefit from a more in-depth analysis of customer demographics, behaviors, and preferences related to sub-lease carsharing.

Detection of Potential Sub-lease Customers

Detection Process

While the detection process is outlined, it would be helpful to provide some context on the significance of certain parameters and thresholds chosen, such as the values of Time and Leave. This would add clarity to the decision-making process.

Detection Results

The results are presented effectively, especially focusing on VIP and WeChat Pay customers. However, the paper lacks a critical discussion of potential biases in the detection process and whether the results can be generalized beyond Shenyang.

Business Strategy Implications

The paper rightly emphasizes the importance of tailored strategies for different customer segments. However, it could be more specific in suggesting promotional tactics and campaigns for VIP and WeChat Pay customers.

Conclusions

The conclusions reiterate the main findings but don't offer new insights or implications beyond what has been discussed in the earlier sections. It's an opportunity to highlight the broader implications of the study's findings for sub-lease carsharing as a whole.

Proposals for improvement

While the paper provides valuable insights into sub-lease carsharing and its potential in Shenyang, there are several areas that could be improved:

1. Transparency and Data Sources. Providing details on data sources and collection methods would enhance the paper's credibility. Additionally, disclosing any potential biases in the data or analysis would make the study more robust.

2. In-Depth Customer Analysis. The paper could benefit from a more comprehensive examination of parking customer demographics and behaviors related to sub-lease carsharing. Understanding customer preferences and motivations is crucial for effective promotions.

3. Generalizability. Discussing the potential limitations and generalizability of the study's findings beyond Shenyang would make the research more applicable to a broader audience.

4. Promotional Strategies. While the paper mentions tailoring strategies for VIP and WeChat Pay customers, it could provide more concrete suggestions for promotional tactics and campaigns. This would be especially valuable for stakeholders looking to implement these strategies.

5. Broader Implications. The conclusions section could be enhanced by discussing the broader implications of the study's findings for the sub-lease carsharing industry as a whole. How might these findings inform the development of sub-lease carsharing services in other regions?

Specific comments

Footnotes should be avoided. authors should try to include the data provided in the 3 footnotes on page 6 with appropriate references within the main text.

Figures and tables do not follow Sustainability's template, where the name and number of the figure and table are in bold type. In addition, “Figure x” is followed by a period.

The references also do not seem to follow the sustainability template, where the year appears in boldface type.

In my opinion, the manuscript needs more references to contrast the methodology followed and the results obtained.

Discussion section is mandatory. Please find attached the documentation described in Sustainability template. “Authors should discuss the results and how they can be interpreted from the per-spective of previous studies and of the working hypotheses. The findings and their im-plications should be discussed in the broadest context possible. Future research direc-tions may also be highlighted.”

Author Response

(The authors gave the same response as above.)

Round 2

Reviewer 1 Report

Thanks for authors' carefull response. All comments have been addressed. No further comment is given.

Author Response

We sincerely thank the reviewer for the previous constructive comments.

Reviewer 2 Report

In this second version, the authors have significantly improved the manuscript by following most of the recommendations made. However, the former conclusion section has now been renamed discussion, but it is still a conclusion section where the main contributions of the research are summarized. The authors should prepare a discussion section where the results obtained are discussed and compared with other similar studies carried out in the past.

Author Response

We thank the reviewer for the constructive comment. We have revised our manuscript accordingly. For your convenience, we explain how we have addressed the comment in the attached file. The comment has been marked in italics and our reply is in regular font, while the revised tests in the manuscript are marked in blue.

Round 3

Reviewer 2 Report

In this version, the authors have included a discussion and conclusions section to improve the manuscript.